# CONTRASTIVE PREDICT-AND-SEARCH FOR MIXED INTEGER LINEAR PROGRAMS

## ABSTRACT

Mixed integer linear programs (MILP) are flexible and powerful tools for modeling and solving many difficult real-world combinatorial optimization problems. In this paper, we propose a novel machine learning (ML)-based framework *ConPaS* that learns to predict solutions to MILPs with contrastive learning. For training, we collect high-quality solutions as positive samples and low-quality or infeasible solutions as negative samples. We then learn to make discriminative predictions by contrasting the positive and negative samples. During test time, we predict assignments for a subset of integer variables of a MILP and then solve the resulting reduced MILP to construct high-quality solutions. Empirically, we show that ConPaS achieves state-of-the-art results compared to other ML-based approaches in terms of the quality of and the speed at which the solutions are found.

## 1 INTRODUCTION

Combinatorial optimization (CO) concerns a wide variety of real-world problems, including resource allocation (Manne, 1960), traffic management (Luathep et al., 2011), network design (Huang & Dilkina, 2020) and production planning (Pochet & Wolsey, 2006) problems, and the majority of them are NP-hard problems. Therefore, designing efficient and effective algorithms for CO is important but also challenging. Mixed integer linear programs (MILP) can flexibly encode and solve a broad family of CO and have been popular. It is a mathematical program that optimizes a linear objective subject to linear constraints, with some of the variables constrained to take integer values. Significant research and engineering effort has been dedicated to developing MILP solvers, such as SCIP (Maher et al., 2017), Gurobi (Gurobi Optimization, LLC, 2022) and CPLEX (Cplex, 2009). The backbones of these solvers are Branch-and-Bound (BnB) (Land & Doig, 2010), Branch-and-Cut (Mitchell, 2002) or Branch-Cut-and-Price (Desrosiers & Lübbecke, 2011), which are all optimal tree search algorithms enhanced by a group of heuristics.

In many real-world settings, MILPs from the same application domain often share similar structures and characteristics. The performance of MILP solvers crucially depends on how effective the heuristics are for that application. Recently, there has been an increased interest in data-driven heuristic designs for MILP for various decision-makings in BnB, including which variable to branch on(Khalil et al., 2016; Gasse et al., 2019), which node to expand (He et al., 2014), which cutting plane to add (Paulus et al., 2022), which subset of variables to reoptimize Song et al. (2020) and which heuristic to run next (Khalil et al., 2017; Chmiela et al., 2021).

There is another line of research that focuses on heuristics that generate high-quality solutions to MILPs. In particular, it focuses on generating partial assignments of high-quality feasible solutions. Previously, Nair et al. (2020) propose Neural Diving (ND), where they learn to partially assign values to integer variables and delegate the reduced sub-MILP to a MILP solver, e.g., SCIP. The fraction of variables to assign values to is controlled by a hyperparameter called the coverage rate. A SelectiveNet (Geifman & El-Yaniv, 2019) is trained for each coverage rate that jointly decides which variables to fix and the values to fix to during testing. The main two disadvantages of ND are that (1) enforcing variables to fixed values leads to low-quality or infeasible solutions if the predictions are not accurate enough and (2) it requires training multiple SelectiveNet to obtain the appropriate coverage rate, which is computationally expensive. To mitigate these issues, Han et al. (2022) propose a Predict-and-Search (PaS) framework that deploys a search inspired by the trust region method. Instead of fixing variables, PaS searches for high-quality solutions within a pre-

defined proximity of the predicted partial assignment, which allows better feasibility and finding higher-quality solutions than ND. For both ND and PaS, the effectiveness (i.e., the quality of the solution found) and efficiency (i.e., the speed at which high-quality solutions are found) depend on the accuracy of the machine learning (ML) prediction and the number of variables (controlled by hyperparameters) whose values to fix.

In this paper, we propose a novel ML-based framework *ConPaS*, **Con**trastive **P**redict-**a**nd-**S**earch for MILPs. ConPaS uses contrastive learning to learn to predict (partial) solutions to MILPs. Similar to both ND (Nair et al., 2020) and PaS (Han et al., 2022), we collect a set of optimal and near-optimal solutions as *positive samples*. Different from their approaches, we also collect infeasible and low-quality solutions as *negative samples*. Instead of using a binary cross entropy loss to penalize the inaccurate predictions for each variable separately, we use a contrastive loss that encourages the model to predict solutions that are similar to the positive samples but dissimilar to the negative ones, with similarity measured by dot products (Oord et al., 2018; He et al., 2020).

Empirically, we test ConPaS on a variety of MILP benchmarks, including problems from the NeurIPS Machine Learning for Combinatorial Optimization (ML4CO) competition (Gasse et al., 2022). We show that ConPaS achieves state-of-the-art anytime performance on finding high-quality solutions to MILPs, significantly outperforming other learning-based methods such as ND and PaS in terms of solution quality and speed. In addition, ConPaS shows great generalization performance on test instances that are 50% larger than the training instances.

## 2 BACKGROUND

In this section, we first define mixed integer linear programs and then provide detailed introductions to both Neural Diving (Nair et al., 2020) and Predict-and-Search (Han et al., 2022).

### 2.1 MIXED INTEGER LINEAR PROGRAMMING (MILP)

A *mixed integer linear program (MILP)* $M = (\boldsymbol{A}, \boldsymbol{b}, \boldsymbol{c}, q)$ is defined as

$$\min \boldsymbol{c}^\mathsf{T} \boldsymbol{x} \quad \text{s.t. } \boldsymbol{A}\boldsymbol{x} \leq \boldsymbol{b} \text{ and } \boldsymbol{x} \in \{0,1\}^q \times \mathbb{R}^{n-q}, \tag{1}$$

where $\boldsymbol{x} = (x_1, \ldots, x_n)^\mathsf{T}$ denotes the $q$ binary variables and $n - q$ continuous variables to be optimized, $\boldsymbol{c} \in \mathbb{R}^n$ is the vector of objective coefficients, $\boldsymbol{A} \in \mathbb{R}^{m \times n}$ and $\boldsymbol{b} \in \mathbb{R}^m$ specify $m$ linear constraints. A solution $\boldsymbol{x}$ is *feasible* if its satisfies all the constraints. In this paper, we focus on the mixed-binary formulation above, however, our approach can also handle general integers using the same engineering techniques introduced in Nair et al. (2020).

### 2.2 NEURAL DIVING

Neural Diving (ND) (Nair et al., 2020) learns to generate a Bernoulli distribution for the solution values of binary variables. It learns the conditional distribution of the solution $\boldsymbol{x}$ given a MILP $M = (\boldsymbol{A}, \boldsymbol{b}, \boldsymbol{c}, q)$ defined as $p(\boldsymbol{x}|M) = \frac{\exp(-E(\boldsymbol{x}|M))}{\sum_{x' \in \mathcal{S}_\mathsf{p}^M} \exp(-E(\boldsymbol{x}'|M))}$, where $\mathcal{S}_\mathsf{p}^M$ is a set of optimal or near-optimal solutions to $M$ and $E(\boldsymbol{x}|M)$ is an energy function of a solution $\boldsymbol{x}$ defined as $\boldsymbol{c}^\mathsf{T}\boldsymbol{x}$ if $\boldsymbol{x}$ is feasible or $\infty$ otherwise. ND learns $\boldsymbol{p_\theta}(\boldsymbol{x}|M)$ parameterized by a graph convolutional network to approximate $p(\boldsymbol{x}|M)$ assuming conditional independence between variables $p(\boldsymbol{x}|M) \approx \prod_{i \leq q} p_\theta(x_i|M)$. Since the full prediction $\boldsymbol{p_\theta}(\boldsymbol{x}|M)$ might not give a feasible solution, ND predicts only a partial solution controlled by the coverage rates and employs SelectiveNet (Geifman & El-Yaniv, 2019) to learn which variables' values to predict for each coverage rates. ND uses binary cross-entropy loss combined with the loss function for SelectiveNet to train the neural network. During testing, the input MILP $M$ is then reduced to solving a smaller MILP after fixing the selected variables.

### 2.3 PREDICT-AND-SEARCH

Predict-and-Search (PaS) (Han et al., 2022) uses the same framework as ND to learn to predict $p(\boldsymbol{x}|M)$. Instead of using SelectiveNet to learn to fix variables, PaS searches for near-optimal solutions within a neighborhood based on the prediction. Specifically, given the prediction $p_\theta(x_i|M)$ for

each binary variable, PaS greedily selects $k_0$ binary variables $\mathcal{X}_0$ with the smallest $p_\theta(x_i|M)$ and $k_1$ binary variables $\mathcal{X}_1$ with the largest $p_\theta(x_i|M)$, such that $\mathcal{X}_0$ and $\mathcal{X}_1$ are disjoint ($k_0 + k_1 \leq q$). PaS fixes all variables in $\mathcal{X}_0$ to 0 and $\mathcal{X}_1$ to 1 in the sub-MILP, but also allows $\Delta \geq 0$ of the fixed variables to be flipped when solving it. Formally, let $B(\mathcal{X}_0, \mathcal{X}_1, \Delta) = \{\boldsymbol{x} : \sum_{x_i \in \mathcal{X}_0} x_i + \sum_{x_i \in \mathcal{X}_1} 1 - x_i \leq \Delta\}$ and $D$ be the feasible region of the original MILP, PaS solves the following optimization problem:

$$\min \boldsymbol{c}^\mathsf{T}\boldsymbol{x} \quad \text{s.t. } \boldsymbol{x} \in D \cap B(\mathcal{X}_0, \mathcal{X}_1, \Delta). \tag{2}$$

Restricting the solution space to $B(\mathcal{X}_0, \mathcal{X}_1, \Delta)$ can be seen as a generalization of the fixing strategy employed in ND where $\Delta = 0$. Though in ND, $\mathcal{X}_0$ and $\mathcal{X}_1$ are constructed using sampling methods based on the neural network output.

## 3 RELATED WORK

In this section, we first summarize other related works on solution predictions for CO. We then summarize related works on MILP solving using machine learning and existing contrastive learning methods for solving CO problems.

### 3.1 SOLUTION PREDICTIONS FOR COMBINATORIAL OPTIMIZATION

There are other works on learning to predict solutions to MILPs in addition to ND and PaS. Ding et al. (2020) propose to learn to predict backbone variables (Dubois & Dequen, 2001) whose values stay unchanged across different optimal and near-optimal solutions and then search for optimal solutions based on the predicted backbone variables. However, this approach is not applicable to many domains since backbone variables do not necessarily exist for many CO problems. A recent work (Yoon et al., 2023) proposes threshold-aware learning to optimize the coverage rate in ND and is one of the state-of-the-art approaches in this line of research. However, this approach also fixes variables when solving the sub-MILP. Khalil et al. (2022) and Li et al. (2018) learn to guide decision-making, such as warm-starting and node selection, in CO solvers, such as MIP solvers and local search, via solution predictions.

### 3.2 MACHINE LEARNING-GUIDED MILP SOLVING

Several studies have applied ML to improve BnB for MILP solving. A huge body of such studies focuses on learning to either select variables to branch on (Khalil et al., 2016; Gasse et al., 2019; Gupta et al., 2020; Zarpellon et al., 2021) or select nodes to expand (He et al., 2014; Labassi et al., 2022). There are also a few studies on learning to schedule and run primal heuristics (Khalil et al., 2017; Chmiela et al., 2021) and to select cutting planes (Tang et al., 2020; Paulus et al., 2022; Huang et al., 2022). Large Neighborhood Search (LNS) is a popular heuristic search for MILPs to find high-quality primal solutions quickly. Several learning methods (Song et al., 2020; Sonnerat et al., 2021; Wu et al., 2021; Huang et al., 2023) have been proposed to guide selecting partial solutions to iteratively refine in the search.

### 3.3 CONTRASTIVE LEARNING FOR COMBINATORIAL OPTIMIZATION

Contrastive learning has been studied extensively for visual representations Hjelm et al. (2019); He et al. (2020); Chen et al. (2020) and graph representations You et al. (2020); Tong et al. (2021) but it has not been explored much for solving CO problems. Mulamba et al. (2021) derive a contrastive loss for decision-focused learning to solve CO problems with uncertain inputs that can be learned from historical data, where they view non-optimal solutions as negative samples. Duan et al. (2022) pre-train good representations for the boolean satisfiability problem with contrastive learning.

In a closely related work, Huang et al. (2023) propose CL-LNS that uses contrastive learning to learn heuristics to refine solutions for MILP in LNS. In contrast, ConPaS learns to construct a high-quality (partial) solution from scratch and then find it. ConPaS uses a novel data collection for negative samples and a novel contrastive loss function that considers positive samples with different qualities. While CL-LNS has a limited application to only LNS, the prediction from ConPaS's ML model can be useful in different search algorithms for MILP. First, ConPaS and CL-LNS are complementary to each other and ConPaS can be used to warm start CL-LNS (or any variants of

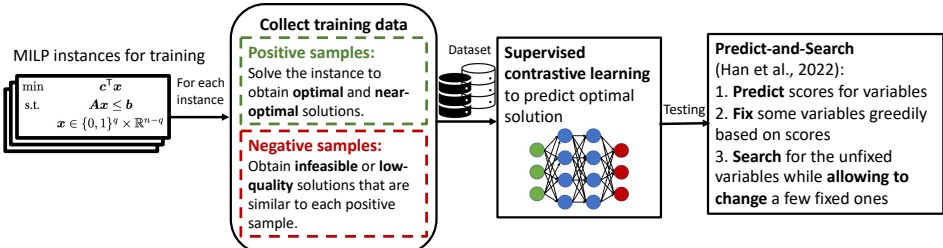

Figure 1: Overview of ConPaS. For training, we collect data from a set of MILP instances, including positive samples that are optimal and near-optimal solutions and negative samples that are low-quality or infeasible solutions. Then, the data is used in supervised contrastive learning to predict optimal solutions. During testing, the predictions are used in Predict-and-Search (Han et al., 2022).

it), similar to what proposed in Sonnerat et al. (2021). One could also leverage the prediction from ConPaS to assign variable branching priorities or generate cuts to improve the performance of BnB.

# 4 SOLUTION PREDICTIONS WITH CONTRASTIVE LEARNING

In this section, we introduce our novel framework *ConPaS*, **Con**trastive **P**redict-**a**nd-**S**earch for MILPs. For a given MILP $M$, our goal is to use contrastive learning to predict the conditional distribution of the solution $p(\boldsymbol{x}|M)$, such that it leads to high-quality solutions fast when it is used to guide downstream MILP solving. In this paper, we mainly focus on using the prediction in Predict-and-Search (optimization problems (2)) following Han et al. (2022). However, such prediction can be used to decompose the feasible regions of the input MILP for exact solving (Ding et al., 2020) or seed LNS with a better primal solution for heuristic solving (Sonnerat et al., 2021). We employ contrastive learning rather than other learning techniques because it has been theoretically demonstrated to be effective (Tian, 2022) and has empirically outperformed alternative approaches in related combinatorial optimization problems (Duan et al., 2022; Huang et al., 2023; Mulamba et al., 2021).

Figure 1 gives an overview of ConPaS. In the following, we describe our novel data collection, our supervised contrastive learning and how we apply solution predictions in the search.

## 4.1 TRAINING DATA COLLECTION

In ConPaS, we use contrastive learning to learn to make discriminative predictions of optimal solutions by contrasting positive and negative samples. Since finding good assignments for integer variables is essentially the most challenging part of solving a MILP, we follow previous work (Nair et al., 2020) to learn $p(\boldsymbol{x}|M)$ approximately as $\prod_{i \leq q} p_\theta(x_i|M)$ where we mainly focus on predicting $p_\theta(x_i|M)$ for binary variables ($i \leq q$). Therefore, our definition of positive and negative samples of solutions mainly concerns the partial solutions on binary variables. Now, we describe how we collect positive and negative samples, which is a crucial step in contrastive learning.

### 4.1.1 POSITIVE SAMPLES COLLECTION

For a given MILP $M$, we collect a set of optimal or near-optimal solutions $\mathcal{S}_{\mathsf{p}}^M$ as our positive samples following previous works (Nair et al., 2020; Han et al., 2022). This is done by solving $M$ exhaustively with a MILP solver and collecting up to $u_{\mathsf{p}}$ best found solutions with the minimum objective values. In experiments, $u_{\mathsf{p}}$ is set to 50.

### 4.1.2 NEGATIVE SAMPLES COLLECTION

Negative samples are critical parts of contrastive learning to help distinguish between high-quality and low-quality (or even infeasible) solutions. For a given MILP $M$, we collect a set of $u_{\mathsf{n}}$ negative samples $\mathcal{S}_{\mathsf{n}}^M$ where $u_{\mathsf{n}} = \beta|\mathcal{S}_{\mathsf{p}}^M|$ and $\beta$ is a hyperparameter to control the ratio between the number

of positive and negative samples. In experiments, $\beta$ is set to 10. We propose two different ways to collect negative samples:

**Infeasible Solutions as Negative Samples.** For each positive sample $\boldsymbol{x} \in \mathcal{S}_{\mathsf{p}}^M$, we collect $\beta$ infeasible solutions as negative samples. We randomly perturb 10% of the binary variable values in $\boldsymbol{x}$ (i.e., flipping from 0 to 1 or 1 to 0). If the MILP $M$ contains only binary variables, we validate that the perturbed solutions are indeed infeasible if they violate at least one constraint in $M$. If $M$ contains both binary and continuous variables, we fix the binary variables to the values in the perturbed solutions and ensure that no feasible assignment of the continuous variables exists using a MILP solver. If less than $\beta$ negative samples are found after validating $2\beta$ perturbed samples, we increase the perturbation rate by 5% and repeat the same process until we have $\beta$ samples.

**Low-Quality Solutions as Negative Samples.** For each positive sample $\boldsymbol{x} = (x_1, \ldots, x_n) \in \mathcal{S}_{\mathsf{p}}^M$, we find the worst $\beta$ feasible solutions that differ from $\boldsymbol{x}$ in at most 10% of the binary variables. If the MILP $M = (\boldsymbol{A}, \boldsymbol{b}, \boldsymbol{c}, q)$ contains only binary variables, we find negative samples $\boldsymbol{x}'$ by solving the following Local Branching (LB) (Fischetti & Lodi, 2003) MILP:

$$\max \boldsymbol{c}^\mathsf{T} \boldsymbol{x}'$$
$$\text{s.t.} \quad \boldsymbol{A}\boldsymbol{x}' \leq \boldsymbol{b}, \boldsymbol{x}' \in \{0,1\}^q \times \mathbb{R}^{n-q}, \tag{3}$$
$$\textstyle\sum_{i \leq q:x_i=0} x_i' + \sum_{i \leq q:x_i=1}(1 - x_i') \leq k.$$

The above MILP is essentially solving the same problem as $M$, but with a negated objective function that tries to find solution $x'$ as low-quality as possible and a constraint that allows changing at most $k$ of the binary variables. After solving it, we consider only solutions as negative samples if they are worse than a given threshold. $k$ is initially set to $10\% \times q$, but if less than $\beta$ negative samples are found with the current $k$, we increase it by 5% and resolve optimization problem (3). We repeat the same process until we have $\beta$ negative samples.

If $M$ contains continuous variables, the goal is to find partial solutions on binary variables, such that we get as low-quality solutions $\boldsymbol{x}'$ as possible when we fix the binary values and optimize for the rest of the continuous variables in $M$. Formally, solving for the partial solutions on binary variables $x_1', \ldots, x_q'$ can be written as a maximin optimization problem:

$$\max_{x_1', \ldots, x_q'} \min_{x_{q+1}', \ldots, x_n'} \boldsymbol{c}^\mathsf{T} \boldsymbol{x}'$$
$$\text{s.t.} \quad \boldsymbol{A}\boldsymbol{x}' \leq \boldsymbol{b}, \boldsymbol{x}' \in \{0,1\}^q \times \mathbb{R}^{n-q}, \tag{4}$$
$$\textstyle\sum_{i \leq q:x_i=0} x_i' + \sum_{i \leq q:x_i=1}(1 - x_i') \leq k.$$

Solving the above maximin optimization exactly is prohibitively hard and, to the best of our knowledge, there are no general-purpose solvers for it (Beck & Schmidt, 2021, Chapter 7). Therefore, we use a heuristic approach where we iteratively solve the inner minimization problem and add a constraint $\boldsymbol{c}^\mathsf{T} \boldsymbol{x}' > \boldsymbol{c}^\mathsf{T} \boldsymbol{x}^*$ to enforce the next solution found is strictly better than the current best-found solution $\boldsymbol{x}^*$ to the maximin problem. It terminates until no better solution can be found. For faster convergence, we sometimes enforce the next solution found to be at least $\epsilon > 0$ better than $\boldsymbol{x}^*$, i.e., we add $\boldsymbol{c}^\mathsf{T} \boldsymbol{x}' \geq \boldsymbol{c}^\mathsf{T} \boldsymbol{x}^* + \epsilon$, where $\epsilon$ is a hyperparameter tuned adaptively in a binary search manner. If we find less than $\beta$ samples, we adjust $k$ the same way as in the previous case.

## 4.2 Supervised Contrastive Learning

### 4.2.1 Neural Network Architecture

Following previous work on learning for MILPs (Gasse et al., 2019), we use a bipartite graph representation to encode the input MILP $M = (\boldsymbol{A}, \boldsymbol{b}, \boldsymbol{c}, q)$. The bipartite graph consists of $n + m$ nodes representing the $n$ variables and $m$ constraints on two sides, respectively, with an edge connecting a variable and a constraint if the variable has a non-zero coefficient in the constraint. Following Nair et al. (2020) and Han et al. (2022), we use features proposed by Gasse et al. (2019) for node features and edge features in the bipartite graph. We learn $\boldsymbol{p_\theta}(\boldsymbol{x}|M)$ represented by a graph convolutional network (GCN) parameterized by learnable weights $\boldsymbol{\theta}$. The GCN takes the bipartite graph representation of $M$ and the features as input. Following Gasse et al. (2019), we perform two rounds of message passing through the GCN to obtain an embedding of the variables, which is then passed through a multi-layer perceptron (MLP) followed by a sigmoid activation layer to obtain the final output $p_\theta(x_i|M)$. Details of the GCN architecture are included in Appendix.

### 4.2.2 TRAINING WITH A CONTRASTIVE LOSS

Given a set of MILP instances $\mathcal{M}$ for training, let $\mathcal{D} = \{(\mathcal{S}_{\mathsf{p}}^M, \mathcal{S}_{\mathsf{n}}^M) : M \in \mathcal{M}\}$ be the set of positive and negative samples for all training instances. A contrastive loss is a function whose value is low when the predicted $p_{\boldsymbol{\theta}}(\boldsymbol{x}|M)$ is similar to the positive samples $\mathcal{S}_{\mathsf{p}}^M$ and dissimilar to the negative samples $\mathcal{S}_{\mathsf{n}}^M$. With similarity measured by dot products, we use an alternative form of InfoNCE (Oord et al., 2018; He et al., 2020), a supervised contrastive loss, that takes into account the solution qualities of both positive and negative samples:

$$\mathcal{L}(\boldsymbol{\theta}) = \sum_{(\mathcal{S}_{\mathsf{p}}^M, \mathcal{S}_{\mathsf{n}}^M) \in \mathcal{D}} \frac{-1}{|\mathcal{S}_{\mathsf{p}}^M|} \sum_{\boldsymbol{x}_{\mathsf{p}} \in \mathcal{S}_{\mathsf{p}}^M} \log \frac{\exp(\boldsymbol{x}_{\mathsf{p}}^{\mathsf{T}} p_{\boldsymbol{\theta}}(\boldsymbol{x}|M)/\tau(\boldsymbol{x}_{\mathsf{p}}|M))}{\sum_{\boldsymbol{x}' \in \mathcal{S}_{\mathsf{n}}^M \cup \{\boldsymbol{x}_{\mathsf{p}}\}} \exp(\boldsymbol{x}'^{\mathsf{T}} p_{\boldsymbol{\theta}}(\boldsymbol{x}|M)/\tau(\boldsymbol{x}'|M))}$$

where we let $\frac{1}{\tau(\boldsymbol{x}|M)} \propto -E(\boldsymbol{x}|M)$ if $\boldsymbol{x}$ is feasible to $M$ where $E(\boldsymbol{x}|M)$ is the same energy function used in previous works (Han et al., 2022; Nair et al., 2020); otherwise $\tau(\boldsymbol{x}|M)$ is set to a constant $\tau$ ($\tau = 1$ in experiments). Intuitively, setting $\tau(\boldsymbol{x}|M)$ in this manner encourages the predictions $p_{\boldsymbol{\theta}}(\boldsymbol{x}|M)$ to be more similar to positive samples $\boldsymbol{x}_{\mathsf{p}}$ with better objectives.

### 4.3 PREDICT-AND-SEARCH

We apply the predicted solution to reduce the search space of the input MILP using the same method as Predict-and-Search (Han et al., 2022). We greedily select $\mathcal{X}_0$ and $\mathcal{X}_1$ based on the prediction and solve the optimization problem defined by Equation (2) given hyperparameters $k_0, k_1$ and $\Delta$.

## 5 EMPIRICAL EVALUATION

### 5.1 SETUP

**Benchmark Problems** We evaluate on four NP-hard benchmark problems that are widely used in existing studies (Gasse et al., 2019; Han et al., 2022), which consist of two graph optimization problems, namely the minimum vertex cover (MVC) and maximum independent set (MIS) problems, and two non-graph optimization problems, namely the combinatorial auction (CA) and item placement (IP) problems. Both MVC and MIS instances are generated according to the Barabasi-Albert random graph model (Albert & Barabási, 2002), with 6,000 nodes and an average degree of 5 following. CA instances are generated with 2,000 items and 4,000 bids according to the arbitrary relations in Leyton-Brown et al. (2000). IP instances are taken from the NeurIPS 2021 ML4CO competition Gasse et al. (2022). The competition also uses the workload appointment (WA) problem as another benchmark problem. However, we include the results in the Appendix since our experiments indicate that they are not challenging enough. For each benchmark problem, we have 400, 100 and 100 instances in the training, validation and test sets, respectively. More details of instance generation and the sizes of the instances are included in Appendix.

**Baselines** We compare ConPaS with three baselines: (1) SCIP (v8.0.1) (Maher et al., 2017), the state-of-the-art open-source ILP solver. We allow restart and presolving with the aggressive mode

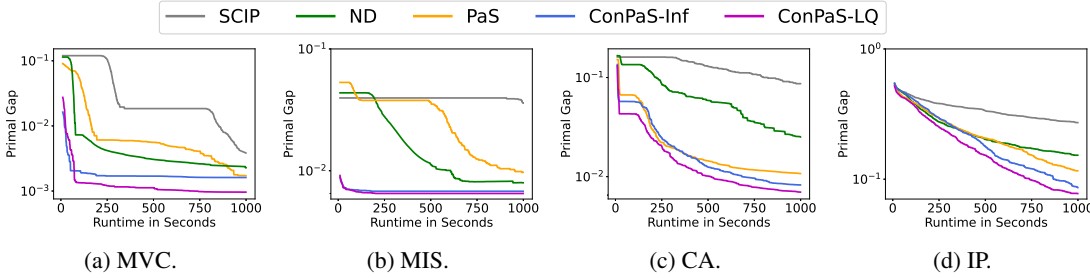

(a) MVC.  (b) MIS.  (c) CA.  (d) IP.

Figure 2: The primal gap (the lower the better) as a function of runtime, averaged over 100 test instances.

turned on for primal heuristics to focus on improving the objective value; (2) Neural Diving (ND) (Nair et al., 2020); and (3) Predict-and-Search (PaS) (Han et al., 2022). We have considered another version of PaS where we replace the neural network output with the LP relaxation solutions of the MILP. However, this approach causes very high infeasibility rates when solving the optimization problem defined by Equation (2). We also compare ConPaS with Gurobi (Gurobi Optimization, LLC, 2022) and present the results in Appendix.

For ML-based approaches, a separate model is trained for each benchmark problem. For PaS, we use the code provided by the authors to train the models. For ND, we implement it and fine-tune its hyperparameters for each problem according to the description in their paper since their code is not available. For PaS and ConPaS, we fine-tune $k_0, k_1$ and $\Delta$ (see definition in Section 2.3) on the validation set before testing.

**Metrics** We use the following metrics to evaluate all approaches: (1) The *primal gap* (Berthold, 2006) is the normalized difference between the primal bound $v$ and a precomputed best known objective value $v^*$, defined as $\frac{|v-v^*|}{\max(v,v^*,\epsilon)}$ if $v$ exists and $v \cdot v^* \geq 0$, or 1 otherwise. We use $\epsilon = 10^{-8}$ to avoid division by zero; $v^*$ is the best primal bound found within 60 minutes by any approach in the portfolio for comparison; (2) The *primal integral* (Achterberg et al., 2012) at runtime cutoff $t$ is the integral on $[0, t]$ of the primal gap as a function of runtime. It captures the quality of the solutions found and the speed at which they are found; and (3) The *survival rate* to meet a certain primal gap threshold is the fraction of instances with primal gaps below the threshold Sonnerat et al. (2021).

**Hyperparameters** We conduct experiments on 2.4 GHz Intel Core i7 CPU with 16 GB memory. Training is done on a NVIDIA P100 GPU with 32 GB memory. All experiments use the hyperparameters described below unless stated otherwise. For data collection, we collect 50 best found solutions for each training instance with an hour runtime using Gurobi (v10.0.0) (Gurobi Optimization, LLC, 2022).For training, we use the Adam optimizer (Kingma & Ba, 2015) with learning rate $10^{-3}$. We use a batch size of 8 and train for 100 epochs (the training typically converges in less than 50 epochs and 5 hours). For testing, we set the runtime cutoff to 1,000 seconds to solve the reduced MILP of each test instance with SCIP (v8.0.1).To tune $(k_0, k_1, \Delta)$ for both PaS and ConPaS, we first fix $\Delta = 5$ or 10 and vary $k_0, k_1$ to be $0\%, 10\%, \ldots, 50\%$ of the number of binary variables to test

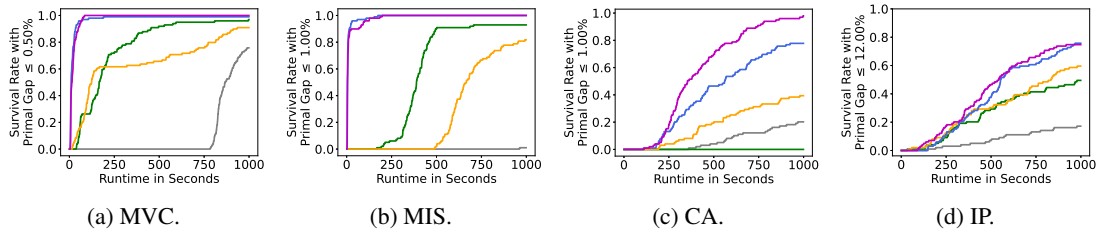

| (a) MVC. | (b) MIS. | (c) CA. | (d) IP. |

Figure 3: The survival rate (the higher the better) to meet a certain primal gap threshold over 100 test instances as a function of runtime. The primal gap threshold is set to the medium of the average primal gaps at 1,000 seconds runtime cutoff among all approaches rounded to the nearest 0.50%.

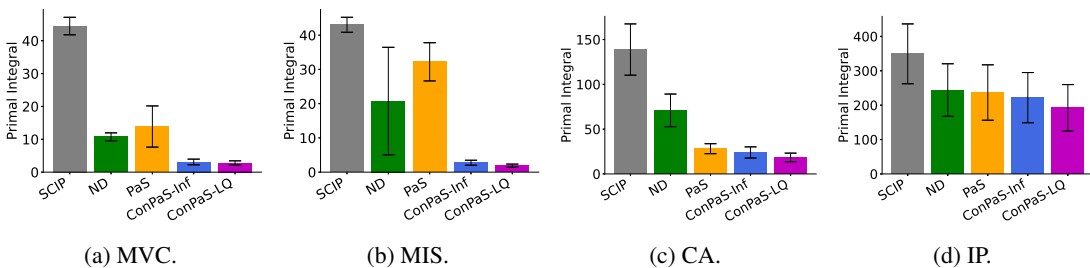

| (a) MVC. | (b) MIS. | (c) CA. | (d) IP. |

Figure 4: The primal integral (the lower the better) at 1,000 seconds runtime cutoff, averaged over 100 test instances. The error bars represent the standard deviation.

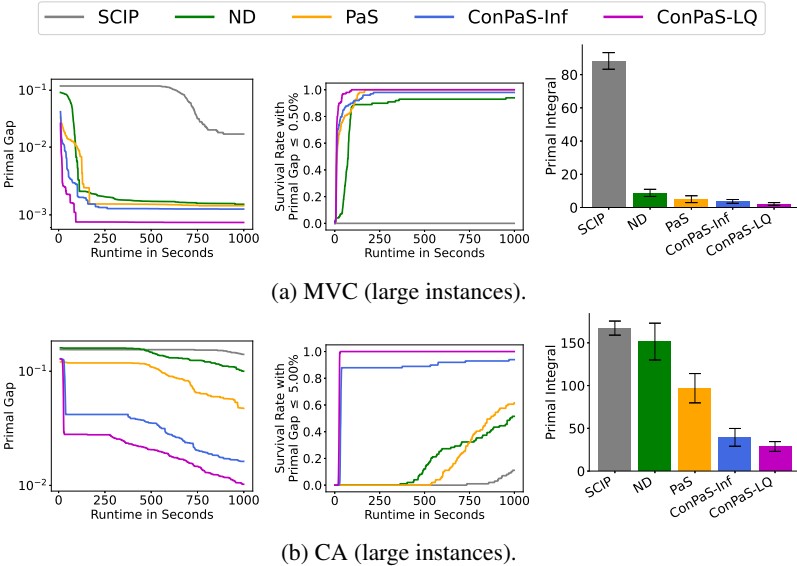

(a) MVC (large instances).

(b) CA (large instances).

Figure 5: Generalization to 100 large instances: The primal gap as a function of runtime, the survival rate as a function of runtime and the primal integral at 1,000 seconds runtime cutoff. The primal gap threshold for the survival rate is chosen as the medium of the average primal gaps at 1,000 seconds runtime cutoff among all approaches rounded to the nearest 0.50%.

their performance on the validation set to get their initial values. We then adjust $\Delta, k_0, k_1$ around their initial values to find the best ones. The values used are reported in Appendix.

## 5.2 RESULTS

We test two variants of ConPaS, denoted by ConPaS-Inf and ConPaS-LQ, that use infeasible solutions and low-quality solutions as negative samples, respectively. Figure 2 shows the primal gap as a function of runtime. Overall, SCIP performs the worst. PaS achieves lower average primal gaps than ND on three of the problems at 1,000 seconds runtime cutoff. Both ConPaS-Inf and ConPaS-LQ show significantly better anytime performance than all baselines on all benchmark problems. ConPaS-LQ performances slightly better than ConPaS-Inf. At the 1,000 seconds runtime cutoff, ConPaS-Inf achieves 3.54%-52.83% lower average primal gaps and ConPaS-LQ achieves 9.82%-86.02% lower average primal gaps than the best baseline.

Figure 3 shows the survival rate to meet a certain primal gap threshold. The primal gap threshold is chosen as the medium of the average primal gap at 1,000 seconds runtime cutoff among all approaches rounded to the nearest 0.50%. ND surprisingly has the lowest survival rate (even lower than SCIP) on the CA instances, indicating high variance in performance of both SCIP and ND[1], but ND is better than both SCIP and PaS on both the two graph optimization problems. PaS has higher survival rates on the CA and IP instances. ConPaS-Inf and ConPaS-LQ have the best survival rate at 1,000 minutes runtime cutoff on all instances. Specifically, on the MVC and MIS instances, at the runtime cutoffs when they both first reach 100% survival rates, the best baseline only achieves about 10%-80% survival rates. These results indicate that ConPaS not only finds better solutions on average, but also finds them for more instances. Figure 4 shows the average primal integral at 1,000 seconds runtime cutoff. The result demonstrates that both ConPaS-Inf and ConPaS-LQ not only find better solutions than the other approaches, but also find them at a faster speed.

### 5.2.1 GENERALIZATION TO LARGER INSTANCES

We test the generalization performance of the trained models on larger instances. We generate 100 large MVC instances according to the Barabasi-Albert random graph model Albert & Barabási

---

[1]When the primal gap threshold is set to 5.00%, ND has a 98% survival rate whereas SCIP has only 56%

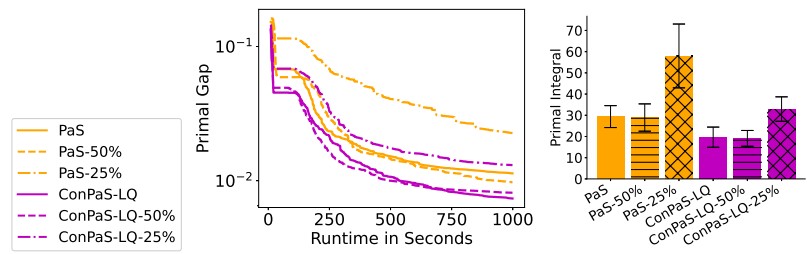

Figure 6: Training on different fractions of training instances: The primal gap as a function of runtime and the primal integral at 1,000 seconds runtime cutoff. ConPaS-LQ-50% and ConPaS-LQ-25% denote the versions of ConPaS trained with only 50% and 25% of the training instances, respectively (similarly for PaS).

(2002), with 9,000 nodes and an average degree of 5. We also generate 100 large CA instances with 3,000 items and 6,000 bids according to the arbitrary relations in Leyton-Brown et al. (2000). These larger instances have 50% more variables and constraints than the previous test instances.

In Figure 5, we show the results of the average primal gaps, survival rates and the average primal integral over 100 test instances. All ML-based approaches demonstrate good generalizability. On large MVC instances, ND, PaS and ConPaS-Inf perform similarly in terms of the primal gap, while ConPaS-Inf improves the primal gap faster than the other approaches. On large CA instances, both ConPaS-Inf and ConPaS-LQ are significantly better than the other baselines in terms of all performance metrics. Overall, on both large MVC and CA instances, ConPaS-LQ is the best and its primal integral at 1,000 seconds runtime cutoff is 57.9%-70.3% lower than the best baseline PaS. It also reaches 100% survival rates fastest for the given thresholds.

### 5.2.2 THE EFFECT OF HYPERPARAMETERS

Next, we study the effect of hyperparameters. Specifically, we focus our study on PaS and ConPaS-LQ on the CA instances. We first empirically study how many training instances are needed for each approach. We train separate models with 50% and 25% of the training instances and test their performance on the test instances. Figure 6 shows the results on the primal gap and primal integral. The two models for ConPaS-LQ trained with 50% and 100% of the instances perform similarly to each other. This is also true for PaS, but its two models are both worse than ConPaS-LQ. When we use 25% of the training instances, we observe a drop in performance for both approaches. However, in this case, ConPaS-LQ performs much better than PaS and only slightly worse than PaS trained on 100% or 50% instances. These empirical results indicate that contrastive learning can achieve better performance using fewer training instances than other learning approaches. In Appendix, we also study how sensitive the performances of ConPaS and PaS are to different $(k_0, k_1, \Delta)$.

## 6 CONCLUSION

We proposed ConPaS, a contrastive predict-and-search framework for MILPs. We learned to predict high-quality solutions by contrasting optimal and near-optimal solutions with infeasible or low-quality solutions. In testing, we solved a reduced MILP by restricting the search space to proximity to the predicted solutions. In experiments, we showed that ConPaS found solutions not only better but also faster than the baselines, which include two state-of-the-art ML-based approaches. ConPaS also demonstrated generalizability to larger instances that were unseen during training. Solving MILPs based on solution predictions, such as ConPaS, ND and PaS, does not guarantee completeness or optimality. The contrastive-learned model in ConPaS can be used in different ways, e.g., to set branching priority in Branch-and-Cut. We believe it is important and interesting for future work to integrate it into optimal tree searches.

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

# APPENDIX

## A  GCN ARCHITECTURE

We follow previous work (Gasse et al., 2019; Han et al., 2022) to use a bipartite graph representation to encode a MILP $M$. For the node (variable and constraint)and edge features of the bipartite graph, we use the same features as Han et al. (2022).

We use the same GCN architecture as previous work (Han et al., 2022). The GCN takes as input the bipartite graph representation of a MILP $M$ with its features and outputs $\boldsymbol{p_\theta}(\boldsymbol{x}|M)$, a $[0,1]$-score vector for the binary variables. For node features, we use 2-layer multi-layer perceptrons (MLP) with 64 hidden units per layer and ReLU as the activation function to map them to $\mathbb{R}^{64}$. We then perform two rounds of message-passings, the first one from variable nodes to constraint nodes and the second one from constraint nodes to variable nodes ,using graph convolution layers (Gasse et al., 2019) to obtain a final variable embedding. The final variable embedding is then passed through a 2-layer MLP with 64 hidden units per layer and ReLU as the activation function followed by a sigmoid layer to obtain the output $\boldsymbol{p_\theta}(\boldsymbol{x}|M)$.

## B  BENCHMARK PROBLEM DESCRIPTIONS AND MILP FORMULATIONS

We present the problem descriptions and MILP formulations for the minimum vertex cover (MVC), maximum independent set (MIS) and combinatorial auction (CA) problems. The descriptions and formulations for the item placement and workload appointment problems could be found at the ML4CO competition Gasse et al. (2022) website[2].

In the MVC problem, given an undirected graph $G = (V, E)$ with a weight $w_v$ associated with each node $v \in V$, we want to select a subset of nodes $V' \subseteq V$ with the minimum sum of weights such that at least one end point of the edge is selected in $V'$ for any edge in $E$:

$$\min \sum_{v \in V} w_v x_v$$
$$\text{s.t.} \quad x_u + x_v \geq 1, \, \forall (u, v) \in E,$$
$$x_v \in \{0, 1\}, \, \forall v \in V.$$

In the MIS problem, given an undirected graph $G = (V, E)$, we want to select the largest subset of nodes $V' \subseteq V$ such that no two nodes in the subsets are connected by an edge in $G$:

$$\min -\sum_{v \in V} x_v$$
$$\text{s.t.} \quad x_u + x_v \leq 1, \, \forall (u, v) \in E,$$
$$x_v \in \{0, 1\}, \, \forall v \in V.$$

In the CA problem, given $n$ bids $\{(B_i, p_i) : i \in [n]\}$ for $m$ items, where $B_i$ is a subset of items and $p_i$ is the bidding price for $B_i$, we want to allocate items to bids such that the total revenue is maximized:

$$\min -\sum_{i \in [n]} p_i x_i$$
$$\text{s.t.} \quad \sum_{i:j \in B_i} x_i \leq 1, \, \forall j \in [m],$$
$$x_i \in \{0, 1\}, \, \forall i \in [n].$$

## C  HYPERPARAMETERS

In this section, we discuss the hyperparameters used for ND, PaS and ConPaS.

For ND, following Nair et al. (2020), we train a model separately for each coverage rate values. Due to limited computing resources, we train models with coverage rate values in $\{0.2, 0.3, 0.4\}$.

---

[2]ML4CO Competition Website: `https://github.com/ds4dm/ml4co-competition/blob/main/DATA.md`

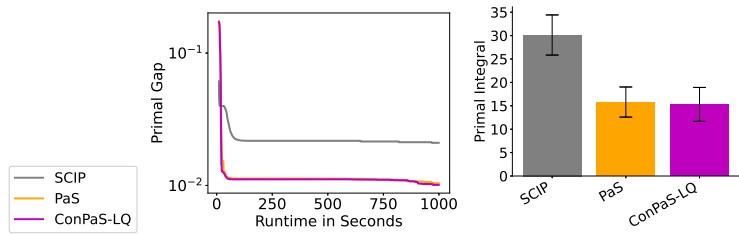

Figure 7: The primal gap as a function of runtime and the primal integral at 1,000 seconds runtime cutoff. Note that the curves of PaS and ConPaS highly overlap with each other.

Table 1: The average numbers of variables and constraints in the test instances.

| Benchmark Problem | MVC | MIS | CA | IP |
|---|---|---|---|---|
| #Binary Variables | 6,000 | 6,000 | 4,000 | 1,050 |
| #Continuous Variables | 0 | 0 | 0 | 33 |
| #Constraints | 29,975 | 29,975 | 2,675 | 195 |

The best coverage rates we found for the MVC, MIS, CA and IP problems are 0.2, 0.2, 0.4 and 0.3, respectively.

For PaS and ConPaS, the values of $k_0, k_1$ and $\Delta$ are summarized in Table 2. Note that the best hyperparameters for both MVC and MIS are quite different for PaS and ConPaS. On MVC instances for PaS, we observe that $(k_0, k_1, \Delta) = (600, 200, 20)$ has a smaller primal integral than $(500, 100, 10)$ but has a larger primal gap at 1,000 seconds runtime cutoff. We also test $(k_0, k_1, \Delta) = (500, 100, 10)$ for ConPaS-LQ, it converges to the same primal gaps (with $< 0.002\%$ differences) as $(800, 200, 20)$ but has a 34.1% increase in primal integral. On MIS instances for PaS, we observe that increasing $k_0$ or $\Delta$ (or both) leads to significantly worse performance. However, if we use $(k_0, k_1, \Delta) = (600, 600, 6)$ for ConPaS-LQ, it converges to the same primal gaps (with $< 0.032\%$ differences) as $(1000, 600, 15)$ but has a 131.8% increase in primal integral (still being better than any other baseline).

## D  ADDITIONAL EXPERIMENTAL RESULTS

### D.1  RESULTS ON THE WORKLOAD APPOINTMENT PROBLEM

Figure 7 presents the results on the WA instances. Both PaS and ConPaS-LQ outperform SCIP significantly in terms of the primal gap and the primal integral. However, both approaches converge quickly to low primal gaps, with ConPaS-LQ being very slightly better than PaS.

### D.2  COMPARISONS WITH GUROBI

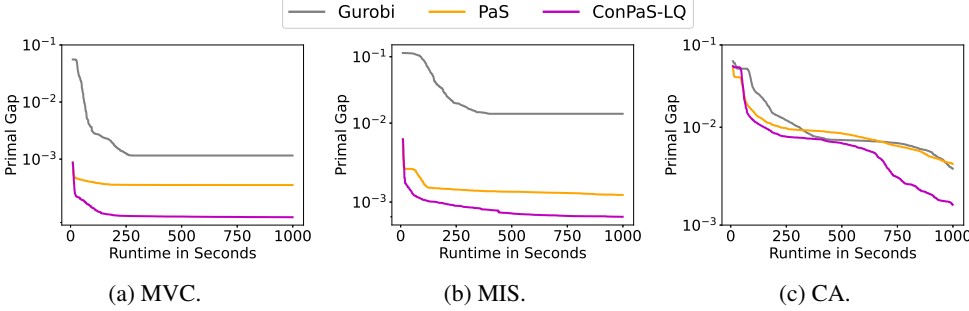

(a) MVC.

(b) MIS.

(c) CA.

Figure 8: Comparisons with Gurobi: The primal gap (the lower the better) as a function of runtime, averaged over 100 test instances.

Table 2: Hyperparameters $(k_0, k_1, \Delta)$ used for PaS and ConPaS.

|      | PaS | ConPaS-Inf | ConPaS-LQ |
|------|-----|------------|-----------|
| MVC  | $(500, 100, 10)$  | $(800, 200, 20)$   | $(800, 200, 20)$   |
| MIS  | $(600, 600, 5)$   | $(1200, 600, 10)$  | $(1000, 600, 15)$  |
| CA   | $(2000, 0, 0)$    | $(2000, 0, 0)$     | $(2000, 0, 0)$     |
| IP   | $(400, 5, 3)$     | $(400, 5, 5)$      | $(400, 5, 2)$      |

Table 3: The primal gap and primal integral at 1,000 seconds runtime cutoff on the CA instances with different $k_0$, averaged over 100 instances.

|        | Primal Gap (%) | | Primal Integral | |
|--------|-----|-----------|------|-----------|
| $k_0$  | PaS | ConPaS-LQ | PaS  | ConPaS-LQ |
| 800    | 6.28  | 6.59    | 114.4 | 117.5 |
| 1200   | 5.45  | 5.05    | 104.3 | 97.3  |
| 1600   | 2.91  | 2.06    | 75.6  | 70.4  |
| 2000   | 1.17  | **0.55** | 28.9 | **19.7** |
| 2400   | 2.19  | 1.40    | 27.5  | 22.9  |
| 2700   | 5.63  | 4.58    | 58.0  | 47.4  |
| 3000   | 12.74 | 11.56   | 127.8 | 115.8 |

We compare the performance of ConPaS-LQ against PaS and Gurobi on the MVC, MIS and CA instances. Note that in this experiment, we use Gurobi in the Predict-and-Search phase for both PaS and ConPaS-LQ to ensure a fair comparison. The hyperparameters $(k_0, k_1, \Delta)$ are reported in Table 5. Figure 8 shows the primal gap as a function of runtime. Figure 9 shows the primal integral at 1,000 seconds runtime cutoff. The results show that both PaS and ConPaS-LQ outperform Gurobi significantly on MVC and MIS instances. Overall, ConPaS-LQ is still the best when applied on Gurobi.

## D.3 RESULTS ON THE EFFECT OF HYPERPARAMETERS

We study the effect of different $(k_0, k_1, \Delta)$ on PaS and ConPaS-LQ on the CA instances. For CA instances, fixing both $k_1$ and $\Delta$ to 0 always gives better primal gaps and primal integrals than other values. Therefore, we vary only $k_0$. We present the results on primal gaps and primal integrals in Table 3. Overall, setting $k_0 = 2,000$ gives the best performance for both PaS and ConPaS-LQ. Either increasing or decreasing $k_0$ from 2,000 hurts their performance. However, if we increase

Table 4: The primal integral at 1,000 seconds runtime cutoff, averaged over 100 instances.

|      | SCIP  | ND    | PaS   | ConPaS-Inf | ConPaS-LQ |
|------|-------|-------|-------|------------|-----------|
| MVC  | 44.5  | 10.7  | 13.9  | 3.1        | 2.8       |
| MIS  | 46.3  | 22.9  | 34.5  | 5.5        | 5.4       |
| CA   | 138.9 | 71.0  | 28.9  | 24.0       | 19.7      |
| IP   | 349.3 | 244.0 | 236.8 | 221.8      | 192.0     |

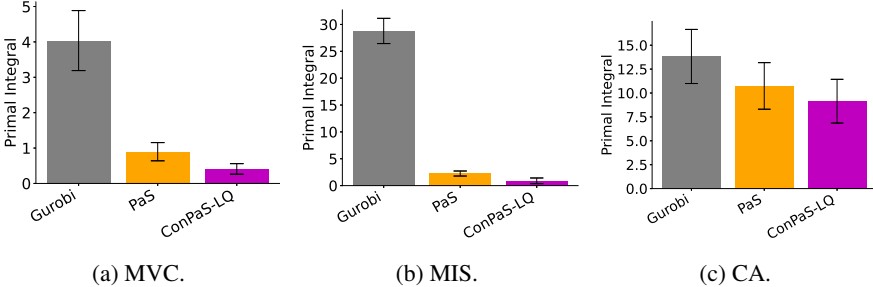

(a) MVC.      (b) MIS.      (c) CA.

Figure 9: Comparisons with Gurobi: The primal integral (the lower the better) at 1,000 seconds runtime cutoff, averaged over 100 test instances. The error bars represent the standard deviation.

Table 5: Comparisons with Gurobi: Hyperparameters $(k_0, k_1, \Delta)$ used for PaS and ConPaS-LQ.

|  | PaS | ConPaS-LQ |
|---|---|---|
| MVC | $(500, 100, 10)$ | $(500, 100, 15)$ |
| MIS | $(500, 500, 10)$ | $(500, 500, 10)$ |
| CA | $(1500, 0, 0)$ | $(1500, 0, 0)$ |

$k_0$ from 2,000, both of them converge to the eventual solutions fast and therefore have comparable primal integrals with small $k_0$, even though sometimes their primal gaps are worse.

In general, having a smaller $k$ requires the search to search for the values on more variables, therefore, it converges slower and has a larger primal integral. On the other hand, having a larger $k$ reduces the search space more, therefore, it converges faster but to a worse solution.

