# OpenReview forum: "Contrastive Predict-and-Search for Mixed Integer Linear Programs"
_ICLR.cc/2024/Conference — Submitted to ICLR 2024_

### Official Review · Reviewer_KTmj · 2023-10-20

**Soundness:** 3 good
**Presentation:** 3 good
**Contribution:** 2 fair
**Rating:** 3
**Confidence:** 4

**Summary:**

In this paper, the authors propose to integrate contrastive learning with the pipeline of solving mixed integer linear programming. They manage to generate positive samples and negative samples during the training. The positive samples are optimal or near-optimal solutions of MILP, while the negative samples are infeasible or low-quality solutions. The model is then trained by these samples via supervised contrastive learning to predict better solutions. After this, the predicted solutions are improved by the PaS framework to make them become valid optimal solutions. Experiments on multiple datasets show the performance of the proposed ConPas framework.

**Strengths:**

1. The paper is well-written and easy to follow.
2. The idea of utilizing contrastive learning in MILP looks interesting to me.
3. The experiments contain various MILP datasets.

**Weaknesses:**

1. I find one work in related work[1] very similar to this paper. Both of these two papers propose to utilize contrastive learning in solving MILP and share the core idea of generating positive and negative samples. The only difference is the operation after the contrastive learning part, the ICML paper[1] uses large neighborhood search (LNS) and this ICLR paper uses Predict and Search (PaS). Actually, I think this paper is covered by the ICML paper, as PaS could be regarded as a variant of LNS. Though the authors do mention this ICML paper in the related work, they do not discuss the difference between their work and the ICML paper, nor compare it as a baseline.
2. Though the idea of utilizing contrastive learning in MILP looks interesting, I consider the current usage of contrastive learning to be more like an incremental part. In this work, solving MILP basically relies on the performance of PaS. I am not sure if this contribution is good enough for ICLR. To me, this work is more like using contrastive learning to find a better initialization for PaS, of which the application is limited.
3. The results of experiments look good, but I think more datasets with hard cases are required. In my own experience of using SCIP,  I think MVS and MIS are relatively easy for SCIP. In contrast, the datasets from NeurIPS 2021 ML4CO are difficult for SCIP, but it looks like the authors did not select the whole datasets of ML4CO, as they said: "IP instances are taken from the NeurIPS 2021 ML4CO competition Gasse et al. (2022)." I wonder how the data is selected. In fact, there are 3 benchmarks in NeurIPS 2021 ML4CO[2], I wonder why the authors neglect them. Besides, a common dataset MIPLIB is also missing in the paper.


[1] Huang, T., Ferber, A.M., Tian, Y., Dilkina, B. &amp; Steiner, B.. (2023). Searching Large Neighborhoods for Integer Linear Programs with Contrastive Learning. <i>Proceedings of the 40th International Conference on Machine Learning</i>, in <i>Proceedings of Machine Learning Research</i> 202:13869-13890 Available from https://proceedings.mlr.press/v202/huang23g.html.

[2] https://www.ecole.ai/2021/ml4co-competition/

**Questions:**

1. Please discuss your paper with the ICML paper I mentioned in the weakness. In my view, these two papers are very similar and the ICML paper seems to cover your work to some extent. A comparison in experiments is also suggested if possible.
2. As I mentioned before, this work is more like using contrastive learning to find a better initialization for PaS. I wonder can this work be applied to methods other than PaS? e.g. Neural Diving mentioned in the paper.
3. The datasets in the experiments require more improvement.

---

> ### Author Response · Authors · 2023-11-19
>
> We thank the reviewer for the feedback and suggestions. Regarding the weaknesses and questions concerning the novelties and choices of MIP benchmark, please kindly refer to the general responses.
>
> In addition, we would like to further discuss how this work could be applied beyond PaS to answer your 2nd question: ConPaS is more versatile since the prediction coming out of its ML model can be useful in different ways. An example is to warm start LNS as mentioned earlier. In addition, one could leverage the ML prediction from ConPaS to assign variable branching priorities and/or generate cuts in tree searches such as branch-and-bound (or branch-and-cut) search. We defer the deployment of ConPaS in different algorithms to future work.
>
> We also want to clarify that Neural Diving is a more restricted variant of ConPaS and PaS, where it corresponds to setting \Delta = 0 in PaS that allows no change of the assigned values once they’re fixed in the search.

---

> ### Comment · Reviewer_KTmj · 2023-11-21
>
> Thank you very much for the detailed response and the improved quality of the paper. However, I think the main concern of this paper is still the difference between ConPas and CL-LNS. I understand that they shall be complementary to each other, but I still believe that the approaches of these two works are similar or at least with strong correlation, as mentioned by other reviewers. Therefore, I think the authors should include the discussion between ConPas and CL-LNS in the **main paper**, instead of just mentioning it as related works, otherwise, it will be suspected of deliberately avoiding. As the authors use a lot of space in the general response to describe the difference in their general response, you can not suppose the readers of your paper understand the difference just by mentioning and citing it. Due to the similarity of ConPas and CL-LNS, It's not an exaggeration to open a separate subsection, which could include a discussion of differences or add a table of comparison. Only in this way, the readers can fully understand the novelty of this work.

---

> > ### Author Response · Authors · 2023-11-21
> >
> > We would like to thank the reviewer for reading our rebuttal and your valuable suggestion on addressing the differences between ConPaS and CL-LNS in the main paper. We agree that it is important for the readers to understand the differences. We have added a paragraph at the end of the related work section highlighted in blue to address this issue. Please kindly let us know if any concerns remain.

---

### Official Review · Reviewer_nvLV · 2023-10-27

**Soundness:** 3 good
**Presentation:** 3 good
**Contribution:** 1 poor
**Rating:** 6
**Confidence:** 3

**Summary:**

The paper presents a method for finding primal solutions to mixed-integer programs using a graph neural network-based approach. The training and performance of the approach is improved through the use of constrastive learning, which has been gaining popularity in a variety of deep reinforcement learning applications due to the fact that it does not require expensive labeling of data to "pre-train" networks. The approach is based on the "predict and search" method from a previous ICLR paper. The approach is evaluated experimentally on several relatively easy MIP problems and a dataset of integer programs from the NeurIPS 2021 ML4CO competition.

**Strengths:**

- Contrastive learning shows great promise in the space of combinatorial optimization; we see again and again that it is an effective mechanism for reducing training time and creating great models.

- The empirical performance of the method on the datasets tested is quite strong.

- (Updated) The novelty of the paper, while not huge, is sufficient for ICLR. The authors have indicated how it differs from CL-LNS, and the bi-level model is an interesting contribution that other groups solving MIPs will want to consider.

**Weaknesses:**

- The instance dataset is not so great, but I admit there are not so many good public MIP problems out there. Simply put, claiming that you can solve the CA dataset to a MIP person is just really not that interesting. Since all the other MIP papers at ICLR/NeurIPS seem to have the same problem, I'll let it pass.

- Using SCIP as a direct point of comparison is not really fair. SCIP is trying to prove optimality, while the method proposed in this work is just a primal heuristic. I appreciate, however, that the authors do not make big claims about beating SCIP the way some papers in this area do. They do seem to understand that beating SCIP is relatively meaningless.

- I am a little surprised to not see an abalation study on the modified loss function. (Update: the authors have provided one, and the modified loss works and is not the only reason it is outperforming previous work)

- The introduction's description of Gurobi and CPLEX is not complete. They are really branch and cut algorithms with (what CPLEX calls) "dynamic search" (and a whole bunch of other stuff, who knows what half of it is...) (Update: this seems to be fixed)

- (Update) I still feel like there could be more experimentation regarding the negative examples (e.g., versus the strategy in the CL-LNS paper??). Since this is the main contribution, I wish it was actually more in focus throughout the paper.

**Questions:**

All questions have been answered.

---

> ### Author Response · Authors · 2023-11-19
>
> We thank the reviewer for the feedback and suggestions. Regarding the weaknesses concerning the novelties, choices of MIP benchmark and using SCIP as a baseline (weaknesses 1,2 and 3), please kindly refer to the general responses to all reviewers posted at the top.
>
> Regarding weaknesses 4 and 5:
>
> (4) We conduct an additional ablation study on ConPaS-LQ on the MVC and CA problems. (Due to limited computation resources, we are still in the process of getting results for ConPaS-inf and other problems.)
> The initial results are shown in the table below, where ConPaS-LQ (unweighted) refers to training using the original InfoNCE function without considering different qualities of the samples and ConPaS-LQ (weighted) refers to training using the modified loss. When we use the original loss function, ConPaS is still able to outperform PaS. Its performance further improves when the modified loss function is used.
>
> |                        | MVC        |                 | CA         |                 |
> |------------------------|------------|-----------------|------------|-----------------|
> |                        | Primal Gap | Primal Integral | Primal Gap | Primal Integral |
> | PaS                    | 0.17%      | 13.9            | 1.16%      | 28.9            |
> | ConPaS-LQ (unweighted) | 0.12%      | 3.3             | 0.57%      | 24.3            |
> | ConPaS-LQ (weighted)   | 0.10%      | 2.8             | 0.16%      | 19.7            |
>
>
>
> (5) We thank the reviewer for the suggestions for a more accurate description for solvers like Gurobi and CPLEX. We have updated the text accordingly in the new draft.

---

> > ### Comment · Reviewer_nvLV · 2023-11-22
> >
> > Dear authors,
> >
> > Thank you for these results. These results are very good to know.
> >
> > Let me emphasize that the statement in bold at the end of your general response does not interest me, and it shouldn't interest the other reviewers either.
> >
> > I now understand the novelty of the paper to be the way you compute the negative examples for the contrastive learning and that there are key differences to CL-LNS. This puts the paper in somewhat of a different light. I don't usually raise scores by this much, but actually the bilevel model for computing negative examples is rather clever and really works well. I encourage the other reviewers to take this into account. I will adjust my review.

---

### Official Review · Reviewer_GFiJ · 2023-10-27

**Soundness:** 2 fair
**Presentation:** 3 good
**Contribution:** 3 good
**Rating:** 5
**Confidence:** 4

**Summary:**

This paper proposes a construction approach of positive and negative samples based on the quality of milp problem’s feasible solutions. With the constructed samples, one can train a GNN model to predict good assignments for integer variables using contrastive learning mechanism, which helps search optimal solution more quickly. Superior experimental results demonstrate the effectiveness and generalizability of the proposed approach.

**Strengths:**

The research topic is valuable and the paper is well written. Moreover, the designed method is presented with succinctly and clearly as well as its motivation. The performance of trained GNN is also impressive, which indicates the superiority of the proposed method.

**Weaknesses:**

There are still some issues needed to be addressed to make this paper meet the requirement of ICLR:
1.	The contribution and novelty is not summarized clearly and relatively weak. The main contributions of this paper are applying contrastive learning to predict and search optimal solution.

2.	Results of empirical evaluation can be more solid and convincing. The experiments are just conducted on two generated dataset and one competition dataset, without the recognized authoritative dataset MIPLIB2017 benchmark. Furthermore, only an open-source MILP solver, which is not well configured, is involved in baselines. Considering that different configuration can significantly affect the solver’s performance, I would expect some further comparative experiments conducted on SCIP configured with tuned parameters or some more powerful commercial solvers (like GUROBI and CPLEX).

**Questions:**

I noticed that the effect of hyperparameter k0 and k1 is evaluated. Of course, this hyperparameter is important, because it controls the tradeoff between the feasibility and quality of predicted solutions. However, considering that MILP instances generally have different scales of integer variables, a specific number of integer variables may not be a good choice. I was wondering that would it be better if we use the coverage rate (i.e., the ratio of fixed variables in the entire set of integer variables when using predict methods like Neural Diving) to control the fixed number of integer variables.

In addition, some studies indicate that each instance has an unique optimal coverage rate (https://arxiv.org/abs/2308.00327), so I think that evaluating the effect of k0 by just computing an average number on one dataset (CA) may not help readers configure their own prediction model properly.

---

> ### Author Response · Authors · 2023-11-19
>
> We thank the reviewer for the feedback and suggestions.
> Regarding the weaknesses, please refer to the discussions on the novelties of the work and choices of benchmark in the general response to all reviewers.
>
> Below are our responses to answer the question regarding hyperparameters and coverage rates:
>
> We agree that using coverage rates as alternatives to model k0 and k1 would be more helpful when the instances are diverse in size. In our paper, we described a systematic way in Section 5.1 “Hyperparameters” to tune both k0 and k1 based on a percentage of the number of variables (10%-50%). We believe that this hyperparameter tuning method is easy to follow. We report the results of different k0 for CA to demonstrate how tuning could be done.
>
> Regarding the optimal coverage rate studied in [Yoon et al., 2023], it is important for methods like Neural Diving (ND) since it requires training a separate model for each coverage rate. With an optimal coverage rate identified, it helps overcome the training inefficiency of ND. However in ConPaS, instead of fixing all variables according to the prediction, we let the MIP solver explore regions around the prediction that allows more flexibility and room for inaccuracy in prediction, therefore removing the need for an accurate coverage threshold.
>
>
> [Yoon et al., 2023] Threshold-aware Learning to Generate Feasible Solutions for Mixed Integer Programs. Arxiv 2023.

---

### Official Review · Reviewer_fyvF · 2023-11-01

**Soundness:** 3 good
**Presentation:** 3 good
**Contribution:** 2 fair
**Rating:** 5
**Confidence:** 4

**Summary:**

The authors propose a predict-and-search approach for solving mixed integer programming(MIP), according to a GNN-guided approach from [Han2022]. The algorithm collects high-quality solutions as positive samples and low-quality solutions as negative samples, and then trains the prediction model by contrastive learning. The authors demonstrate that the proposed method outperforms the baseline on four commonly used mixed-integer linear programming datasets.

**Strengths:**

1. The effect of improving the prediction model through contrastive learning training method is intuitive and effective.
2. The author's experiments show that the proposed method has a significant improvement over the baseline.
3. The paper is mostly well-written and easy to follow.

**Weaknesses:**

1. The technical novelty is limited. First, it is a somewhat straightforward application of contrastive learning to predict-and-search. Second, the proposed method is essentially the same as the ICML 2023 paper [Huang2023] (Figure 1 of this paper almost coincides with Figure 1 in [Huang2023]), if we consider the procedure as a one-step LNS.

2. Since the proposed approach is based on predict-and-search, it cannot guarantee the optimality or feasibility. This limitation is not discussed or analyzed properly in this paper. For example, there is no empirical study on the feasibility ratio on the test instances. The authors should also conduct experiments on more constrained problems. Furthermore, it is somewhat unfair to compare the anytime performance with SCIP, since the proposed method (as well as predict-and-search) essentially solves a much simpler problem than SCIP since some variables are fixed.

3. The authors collected training data using Gurobi, but only compared the test performance with SCIP. I cannot see any reason why not compare with Gurobi at test time.

4. The authors used two ways to collect negative samples, but only report their empirical performance, without a deep analysis on which way is more reasonable.

5. The authors did not show the results of how accurate the solution prediction is.

**Questions:**

Please see the above weaknesses.

---

> ### Author Response · Authors · 2023-11-19
>
> We thank the reviewer for the feedback and suggestions.
> Regarding the weaknesses in the novelties and comparison to SCIP and Gurobi (weaknesses 1,2 and 3), please kindly refer to the general responses to all reviewers.
>
> To address the other weaknesses:
>
> 2. ConPaS is a solution construction heuristic and we acknowledge that our approach doesn’t guarantee optimality or feasibility. Similarly, this drawback also applies to Neural Diving (ND) [Nair et al, 2020] and PaS [Han et al, 2023]. However, in a distributional setting where one needs to solve similar MIP instances over and over again, approaches like ND and ConPaS can be particularly helpful if it is able to predict solutions that are empirically feasible and of high quality. This is indeed true according to our experiments - on the five MIP benchmarks (including one in the Appendix), we achieve a 100% feasibility rate using a consistent set of hyperparameters on each benchmark, confirming the applicability of these approaches. However, we also acknowledge that ConPaS (or ND and PaS) is not universally applicable to all MIP solving, especially on more constrained problems. For example, using MIP for scientific discoveries when the solutions are sparse could be extra challenging [Deza et al., 2023] and often we need to design other approaches tailored to them. We have added the discussion in the conclusion section.
>
> 3. Thank you for this comment. We use Gurobi to collect data since Gurobi typically runs a lot faster than SCIP. For data collection, we set the time limit to an hour for Gurobi. We could easily replace Gurobi with SCIP for data collection and get the same-quality training data but this comes at the cost of 4-8 times (4-8 hours per instance)  longer runtime on average. Due to our limited computational resources, using Gurobi for data collection is more practical for us.
>
> We have included results on Gurobi in Appendix Section D.2 in the updated draft. We show that ConPaS still outperforms Gurobi and PaS significantly in terms of both primal integral and primal gap.
>
>
> 4. The main motivation to design negative samples this way is that we want them to be close to positive samples in the input space but actually with very different quality (i.e., near miss). From a theoretical point of view, the InfoNCE loss we use has the property that it will automatically focus on hard negative pairs (i.e., samples with similar representation but of very different qualities) and learn representations to separate them apart (See e.g., [Tian 2022]). While our approach is built upon a theoretical understanding of contrastive learning, we acknowledge that our work designs the negative samples heuristically and does not aim for theoretical impacts. On the other hand, we believe that our work contributes a new principled method that demonstrates strong empirical performance in challenging domains.
>
>
> 5. Regarding the accuracy of the predicted solutions, we would like to point out that the prediction accuracy doesn’t strongly correlate with the performance of the downstream task where the predictions are used (in this paper, the downstream task is the search phase). The ML model is trained on multiple solution samples and when deployed in the search, we use only a part of the predictions controlled by the hyperparameters. Therefore, there is no standard way to quantify the accuracy of the ML predictions in this setting that captures the downstream performance.
>
> [Nair et al., 2020] Solving mixed integer programs using neural networks, Arxiv 2020.
>
> [Han et al., 2023] A GNN-guided predict-and-search framework for mixed-integer linear programming. ICLR 2023
>
> [Deza et al., 2023] Fast Matrix Multiplication Without Tears: A Constraint Programming Approach. CP 2023
>
> [Tian 2022] Understanding Deep Contrastive Learning via Coordinate-wise Optimization. NeurIPS 2022.

---

> > ### Comment · Reviewer_fyvF · 2023-11-22
> >
> > I thank the authors for giving the detailed response and new experimental results. However, some of my concerns remains.
> >
> > 1. Regarding the novelty over CL-LNS, I think "complementary" is not very sufficient to justify the differences or novelty. Moreover, the authors claimed two novelties, the negative data collection method and new loss function, but they do not provide any ablation study to support their advantages.
> >
> > 2. Regarding the prediction accuracy, I am not satisfied with the response. If the prediction has low impact on the downstream tasks, then why you need a prediction after all? If a poor prediction can also lead to a good final performance, then I suspect the meaning and usefulness of the ML part, and the performance improvement may come from tuning other hyperparameters. So accuracy is important, because it justifies your core contribution which is an ML component. Also, I do not agree with the last statement in response 5. Prediction accuracy is very easy to quantify, and we do not need to involve the downstream tasks here.
> >
> > I will increase my score, but still believe that this paper needs further improvement.

---

> > > ### Author Response · Authors · 2023-11-23
> > >
> > > We thank the reviewer for taking the time to read our rebuttal and the follow-up feedback. Regarding the remaining concerns:
> > >
> > > 1. We propose novel data collection methods to find negative samples that are new and specifically designed for our solution prediction task. Existing methods for finding negative samples such as the one proposed for CL-LNS do not directly apply here.
> > > For the modified contrastive loss function, we conduct an additional ablation study on ConPaS-LQ on the MVC and CA problems.
> > > The initial results are shown in the table below, where ConPaS-LQ (unweighted) refers to training using the original InfoNCE function without considering different qualities of the samples and ConPaS-LQ (weighted) refers to training using the modified loss. When we use the original loss function, ConPaS is still able to outperform PaS. Its performance further improves when the modified loss function is used.
> > >
> > > |                        | MVC        |                 | CA         |                 |
> > > |------------------------|------------|-----------------|------------|-----------------|
> > > |                        | Primal Gap | Primal Integral | Primal Gap | Primal Integral |
> > > | PaS                    | 0.17%      | 13.9            | 1.16%      | 28.9            |
> > > | ConPaS-LQ (unweighted) | 0.12%      | 3.3             | 0.57%      | 24.3            |
> > > | ConPaS-LQ (weighted)   | 0.10%      | 2.8             | 0.16%      | 19.7            |
> > >
> > >
> > > 2. We report the prediction accuracy quantified by the classification accuracy over all binary variables (with the threshold set to 0.5) in the following table. We report it for both PaS and ConPaS-LQ on the MVC and CA problems on 100 validation instances. The accuracy is the fraction of correctly classified variables averaged over 50 positive samples for each instance, and we report the average accuracy over 100 validation instances. Since the classification accuracy is sensitive to the threshold, we also report the AUROC.
> > > On the MVC instances, though ConPaS has a lower accuracy (w.r.t. Threshold =0.5), it has higher AUROC than PaS. On the CA instances, their accuracies and AUROCs are similar. We would like to again point out that a better accuracy/AUROC doesn't necessarily indicate a better downstream task performance, even though we believe they are correlated.
> > >
> > >
> > > |           | MVC      |       | CA       |       |
> > > |-----------|----------|-------|----------|-------|
> > > |           | Accuracy | AUROC | Accuracy | AUROC |
> > > | PaS       | 81.2%    | 0.88  | 88.3%    | 0.87  |
> > > | ConPaS-LQ | 76.9%    | 0.91  | 86.9%    | 0.86  |

---

### Author Response · Authors · 2023-11-19
**General Response**

We are grateful to all reviewers for their time and helpful suggestions. In this general response, we address concerns and answer questions that come from multiple or all reviews. We summarize our responses and additional findings in the rebuttal text, and we encourage reviewers to look at the updated paper draft uploaded to OpenReview. The updated draft contains new experimental results comparing against Gurobi and a few edits to improve the clarity of the paper. The changes are highlighted in blue for visibility.

**Novelties of ConPaS and its Differences from CL-LNS [Huang et al, ICML 2023]**

**Differences**: We would like to clarify that our work ConPaS and existing work CL-LNS published at ICML 2023 this year are complementary to each other. More specifically,
ConPaS learns to construct a high-quality (partial) solution from scratch and then find it,
CL-LNS learns to predict the part of a given solution that is not good enough and then improve it.
One could apply ConPaS to warm start CL-LNS (or any other Large Neighborhood Search (LNS) methods). This is similar to the relationship between Neural Diving (a solution construction method) and the ML-guided LNS (a solution improvement method) demonstrated in [Nair et al., 2020].

Furthermore, while CL-LNS has a limited application to only Large Neighborhood Search, the prediction from ConPaS’s ML model can be useful in different search algorithms for MIP. An example is to warm-start LNS as mentioned above. In addition, one could leverage the ML prediction from ConPaS to assign variable branching priorities and/or generate cuts to improve the performance of tree searches such as branch-and-bound (or branch-and-cut) search. We defer the deployment of ConPaS in those algorithms to future work.

**Novelties**: While both ConPaS and CL-LNS use contrastive learning for MIP solving, we would like to point out our main novelties: (i) We design a novel data collection process with considerations of two types of negative samples. Finding the negative samples is not straightforward especially when using low-quality solutions as negative samples. In that case, we leverage the techniques of local branching (that are more often used to find improved solutions) to find bad solutions that are similar to good ones and formulate it as a nontrivial bilevel optimization problem; (ii) We design a novel contrastive loss function to take into account positive samples with different solution qualities; (iii) We demonstrate strong empirical performance of ConPaS measured by various metrics and we also believe that our work contributes a new and valuable empirical method


**Comparisons with SCIP**

Regarding comparisons with SCIP, we mentioned in the paper that we indeed fine-tuned SCIP heuristic setting in our experiments. Specifically, we set SCIP’s heuristic mode to AGGRESSIVE to focus on primal bound improvement and we also allow presolving and restart heuristics in SCIP. We have made these details clear and highlighted them in the revised draft.
It is a common practice in the MIP-solving community to present SCIP results for completeness. We want to be clear that we do not intend to make big statements about outperforming SCIP, since the main competitors of ConPaS are the other ML-based approaches - ND [Nair et al., 2021] and PaS [Han et al., 2023].

**Comparisons with Gurobi**

We would like to point out that ConPaS is agnostic to the underlying MIP solver that is used in the Predict-and-Search phase. It could be applied to SCIP, Gurobi or CPLEX. In our paper, we demonstrate the effectiveness of ConPaS using SCIP as the solver but it could also be built upon Gurobi. We have included results on Gurobi in Appendix Section D.2 in the updated draft. Due to limited computation resources, we run experiments with Gurobi, PaS [Han et al., 2023] and ConPaS on MVC, MIS and CA instances. **The results show that ConPaS outperforms Gurobi significantly in terms of both the primal gap and primal integral performances.**

---

> ### Author Response · Authors · 2023-11-19
> **General Response 2/2**
>
> **Choice of MIP Problem Benchmark**
>
> We would like to respectfully argue that the benchmark problems used in our paper are already challenging enough for existing MIP solvers such as SCIP and Gurobi, as shown by the results reported in Sections 5.2 and D.2. These benchmarks have indeed been used in various previous studies [Han et al., 2023; Huang et al., 2023; Wu et al., 2021]. We use even larger scale instances for combinatorial auction and independent set problems compared to the closely related recent work [Han et al., 2023].
>
> We would like to clarify that we also use two problem domains (Item placement and workload appointment) from the NeurIPS 2021 ML4CO competition. The results of the workload appointment problems are reported in the Appendix due to being not challenging enough for our setting. We use the same train/validation/test split as suggested by the organizers (we use only 400 instances from their train set though 9,900 instances are given). We would also want to respectfully disagree with the claims that instances from ML4CO competition are harder than the other benchmarks. In the competition, they are indeed hard since the rules of the competition require all heuristics in SCIP (including restart and primal heuristics) to be turned off. However, in our paper, we allow all those options and fine-tune them for our SCIP baseline to maximize its performance. We also found that the workload appointment problem is indeed too easy for approaches like PaS and ConPaS.
>
> We agree with reviewers KTmj and GFiJ that MIPLIB is indeed an important MIP benchmark. However, there are few successful cases of ML-based methods for MIP solving to learn heuristics that can generalize to heterogeneous collections of real-world instances like MIPLIB that are diverse in their sizes, domains and structures. Following the majority of previous work, in our paper, we focus on distributional settings for MIP solving that are also important in real-world applications. However, we believe it is important for future work to develop methods that are generalizable to diverse MIP instances.
>
> [Nair et al., 2020] Solving mixed integer programs using neural networks, Arxiv 2020.
>
> [Han et al., 2023] A GNN-guided predict-and-search framework for mixed-integer linear programming. ICLR 2023
>
> [Huang et al., 2023] Searching Large Neighborhoods for Integer Linear Programs with Contrastive Learning. ICML 2023
>
> [Wu et al., 2021] Learning Large Neighborhood Search Policy for Integer Programming. NeurIPS 2021.

---

### Author Response · Authors · 2023-11-21

Dear reviewers,

Thank you again for taking the time to review our paper. We would be grateful if you could kindly check whether our response has answered your questions, and let us know if any issues remain. In addition to our rebuttal response, we have worked to respond to the points raised by the reviewers and submitted a revision.

Here is a summary of our effort to improve the paper draft:

1. We added experimental results comparing the performance with Gurobi where ConPaS still shows significant improvement over the baselines.

2. We added a discussion to the end of Section 3 to address the concerns about the novelties and discuss the differences between CL-LNS and ConPaS.

3. We improved the writing of the paper by improving clarity as well as adding and highlighting some important details.

If you find the responses and revisions align well with the paper's objectives and address your initial concerns, we are hopeful that an adjustment in the score could reflect these improvements. Please feel free to ask if you have more questions or if there's anything else we can provide to support your evaluation.

---

### Meta-Review · Area_Chair_JTNe · 2023-12-06

**Metareview:**

This paper proposes a neural model trained with contrastive learning for solving mixed integer linear programs. It obtains the solution using a recently proposed predict-and-search (PaS) strategy and empirically demonstrates the advantage of using contrastive learning on top of PaS.

All reviewers agree on the good presentation of this paper and the soundness of this work. The experiments on four datasets also show good results. Nonetheless, the novelty of this work is rather limited, as merely an incremental change over PaS (fyvF, GFiJ, KTmj) and the similarity to a recent paper on contrastive learning for large-neighborhood-search (fyvF, KTmj). Also, the lack of the ablation study (fyvF, nvLV) makes the effectiveness of the proposed components questionable.  The authors have addressed part of the concerns during the rebuttal.  However, the overall lack of novelty remains a major issue. Also, a better understanding of the source of performance improvement is needed.

A rejection is recommended.

**Justification For Why Not Higher Score:**

This paper lacks novelty and the ablation study is not well conducted.

**Justification For Why Not Lower Score:**

N/A

---

### Decision · Program_Chairs · 2024-01-16

Reject